# A COMPREHENSIVE OVERHAUL OF DISTILLING UNCONDITIONAL GANS

## ABSTRACT

Generative adversarial networks (GANs) have achieved impressive results on various content generation tasks. Yet, their high demand on storage and computation impedes their deployment on resource-constrained devices. Though several GAN compression methods have been proposed to address the problem, most of them focus on conditional GANs. In this paper, we provide a comprehensive overhaul of distilling unconditional GAN, especially for the popular StyleGAN2 architecture. Our key insight is that the main challenge of unconditional GAN distillation lies in the output discrepancy issue, where the teacher and student model yield different outputs given the same input latent code. Standard knowledge distillation losses typically fail under this heterogeneous distillation scenario. We conduct thorough analysis about the reasons and effects of this discrepancy issue, and identify that the style module plays a vital role in determining semantic information of generated images. Based on this finding, we propose a novel initialization strategy for the student model, which can ensure the output consistency to the maximum extent. To further enhance the semantic consistency between the teacher and student model, we present a latent-direction-based distillation loss that preserves the semantic relations in latent space. Extensive experiments demonstrate the effectiveness of our approach in distilling StyleGAN2, outperforming existing GAN distillation methods by a large margin. Code and models will be released.

## 1 INTRODUCTION

GAN compression (Wang et al., 2020; Li et al., 2020; Liu et al., 2021) has been actively studied to enable the practical deployment of powerful GAN models (Karras et al., 2018; 2019; 2020) on mobile applications and edge devices. Among these techniques, knowledge distillation (Hinton et al., 2015) is a widely adopted training strategy for GAN compression. The objective of GAN distillation is to transfer the rich dark knowledge from the original model (teacher) to the compressed model (student) so as to mitigate the performance gap between these two models. There are two distillation manners, i.e., pixel-level and distribution-level. The former minimizes the distance between generated images of two models, while the latter minimizes the distance between distributions. In this work, we focus on the first setting considering its prevalence in GAN compression literature (Chen et al., 2020; Wang et al., 2020; Li et al., 2020; Liu et al., 2021)

The majority of contemporary GAN distillation methods focus on conditional GANs (cGANs) while the distillation of unconditional GANs (uncGANs) is relatively under-explored. Since there is a large difference between the learning dynamics of these two types of GANs, distillation methods tailored for cGANs cannot be directly applied to the unconditional setting. Hence, in this work, we comprehensively investigate the distilling of uncGANs, especially StyleGAN2 (Karras et al., 2020), considering its popularity in both academia and industry.

We find that the main difficulty of uncGAN distillation lies in the *output discrepancy* between the teacher and student model. In fact, the implicit prerequisite of KD is that teacher and student should have similar outputs for the same input, otherwise the mimicking supervision is no longer meaningful. This prerequisite is easier to be satisfied in cGAN, because the output space of cGANs can be narrowed down by the given conditional input, especially when the condition is strong (Zhu et al., 2017; Isola et al., 2017) Consider horse→zebra task as an example. An input horse image determines which region should be added with zebra stripes and which region is background that should

not be changed. Two generated images in cGAN may differ in some low-level details such as the shape of zebra stripes, but would largely resemble in their structure. Unlike cGAN, as shown in our experiments, it is impossible for a uncGAN student with random initialization to learn similar mapping function to the teacher, even though we leverage distillation loss to enforce the agreement between the outputs of the two models.

To address the aforementioned output discrepancy problem, we carefully examine each component of the StyleGAN2 student model through comparative experiments. We identify that the style module plays a crucial role in deciding the semantic information of the generated images. Based on this finding, we reach to a simple yet effective initialization strategy for the student model, i.e., inheriting the weights from the teacher style module and keeping the remaining convolutional layers randomly initialized. Such initialization strategy can work well even in heterogeneous distillation where the student architecture is obtained by neural architecture search (NAS) or manual design, and is totally different from the teacher model.

After resolving the output discrepancy problem, we further design a more effective mimicking objective tailored for uncGAN distillation. As opposed to most of existing GAN distillation approaches that merely transfer the knowledge within single image, we propose a novel latent-direction-based relational loss to fully exploit the rich relational knowledge between different images. Specifically, we exploit the good linear separability property of StyleGAN2 in latent space and then augment each latent code w by moving it along certain direction such that the resulting image only differs in a *single* semantic factor. Then, we compute the similarity matrix between original images and augmented images and take it as the dark knowledge to be mimicked by the student network. The latent-direction-based augmentation disentangles various semantic factors and makes the learning of each factor easier, thus yielding better distillation performance.

Our **contributions** are summarized as follows: **1)** To the best of our knowledge, this is the first work that uncovers the *output discrepancy* issue in uncGAN distillation. Through carefully designed comparative experiments, we identify that the style module is the determining factor to ensure output consistency. **2)** We propose a concise yet effective initialization strategy for the student model to resolve the output discrepancy problem, demonstrating significant gains upon conventional uncGAN distillation. **3)** We further propose a latent-direction-based distillation loss to employ the rich relational knowledge between different images, and achieve state-of-the-art results in StyleGAN2 distillation, outperforming the existing state-of-the-art CAGAN (Liu et al., 2021) by a large margin.

## 2 RELATED WORK

**GAN Compression.** We highlight a few recent methods among the many GAN compression methods (Shu et al., 2019; Chang & Lu, 2020; Chen et al., 2020; Wang et al., 2020; Li et al., 2020). GAN Slimming (Wang et al., 2020) integrates model distillation, channel pruning and quantization into a unified framework. GAN Compression (Li et al., 2020) first searches a compact student architecture via NAS, and then requires the student to mimic the intermediate outputs and synthesized results of the teacher simultaneously. A common characteristic shared by these works is that they all focus on the compression of cGANs such as Pix2pixGAN (Isola et al., 2017) and CycleGAN (Zhu et al., 2017).

The more recent Content-Aware GAN compression (CAGAN) (Liu et al., 2021) shifts the attention to compressing unconditional GANs. It first estimates the contribution of each channel in each layer to the generated human faces and eliminates channels with little contribution. Subsequently, the pruned model inherits the well-trained parameters from the original network for both style module and convolutional layers, and are finetuned with adversarial loss and distillation loss afterwards. Though CAGAN involves the compression of uncGAN, it bypasses the issues of model heterogeneity between the teacher and student model by allowing the student to inherit the well-trained parameters. Such an requirement assumes the student to inherit the main structure of the teachers too despite pruning. As will be shown in the experiments, the performance of CAGAN will greatly degrade in heterogeneous distillation. The proposed mimicking loss cannot guarantee the student to learn a similar mapping as the teacher. Moreover, we find that the content-aware pruning strategy in CAGAN is not an optimal solution for student initialization. With our proposed initialization strategy, the student model does not need to inherit any weights from convolutional layers of the teacher but achieve better results.

**Knowledge Distillation.** Knowledge distillation (KD) (Hinton et al., 2015) is originally proposed to achieve model compression (Buciluundefined et al., 2006) for image classification, whose target is to transfer the dark knowledge from one or multiple cumbersome networks (teacher) to a small compact network (student). Vanilla KD (Hinton et al., 2015) proposes to match the outputs of two classifiers by minimizing the KL-divergence of the softened output logits. Besides the output logits, other intermediate outputs such as feature maps (Romero et al., 2014), attention maps (Zagoruyko & Komodakis, 2017), Gram matrices (Yim et al., 2017), pre-activations (Heo et al., 2019), relation (Peng et al., 2019; Tung & Mori, 2019) and self-supervision signal (Tian et al., 2020; Xu et al., 2020) can also serve as the dark knowledge. However, it should be careful when adapting KD from classification tasks to generation tasks. The output consistency prerequisite is naturally satisfied in image classification since the supervision of labels guarantees different models to converge to similar mappings. As discussed in Sec. 1, the consistency prerequisite does not naturally hold for uncGANs. Therefore, a special distillation technique tailored for uncGANs is required to cope with the output discrepancy problem.

## 3 METHODOLOGY

### 3.1 PRELIMINARIES

**StyleGAN2.** There are two modules in StyleGAN2 (Karras et al., 2020), i.e., a style module $S(\cdot)$ that maps Gaussian noise $z$ to the style vector $w$ and a convolution backbone $C(\cdot)$ that takes $w$ as input and generate images. The style vector $w$ is fed into the backbone $C(\cdot)$ through the modulated convolution (ModConv) layer (Huang & Belongie, 2017; Karras et al., 2020). And StyleGAN2 allows the use of different $w$ vectors in different ModConv layers. The image generation process can be formulated as:

$$G(z_1, z_2, \cdots, z_L) = C(w_1, w_2, \cdots, w_L) = C(S(z_1), S(z_2), \cdots, S(z_L)), \tag{1}$$

where $L$ is the number of ModConv layers in the backbone, and the $i$-th ModConv layer uses $w_i$ that comes from $z_i$. We define the output consistency condition as:

$$G_s(z_1, z_2, \cdots, z_L) = G_t(z_1, z_2, \cdots, z_L), \tag{2}$$

where the $s$ and $t$ represent student and teacher, respectively. Equation 2 suggests that the generated images of two models should be the same if they use the same $z$ at corresponding layers.

**StyleGAN2 Compression.** A typical StyleGAN2 compression approach (Liu et al., 2021) contains two steps, i.e., pruning and finetuning. In the pruning stage, unimportant / unnecessary channels will be removed according to some heuristics (Hu et al., 2016; Li et al., 2017; He et al., 2018; Liu et al., 2021). Note that pruning is only applied to the convolution backbone $C(\cdot)$ and the style module $S(\cdot)$ is kept *unchanged*. The pruned model will inherit the well-trained weights from the original model for both the style module and the convolution backbone (Liu et al., 2021). In the finetuning stage, besides the normal adversarial loss, the pruned model is also required to mimic the original model's output to compensate the performance degradation brought by channel reduction. A typical mimicking loss includes RGB loss and LPIPS loss (Zhang et al., 2018):

$$\mathcal{L}_{\mathrm{rgb}} = ||G_s(z) - G_t(z)||_1, \mathcal{L}_{\mathrm{lpips}} = ||F(G_s(z)) - F(G_t(z))||_1, \tag{3}$$

where $F$ is a well-trained frozen network that computes the perceptual distance between two images. $L_{\mathrm{rgb}}$ and $L_{\mathrm{lpips}}$ require that the generated image of student should be close to that of teacher in RGB space and perceptual space, respectively. The final loss function in the finetuning stage is:

$$\mathcal{L} = \lambda_{\mathrm{GAN}}\mathcal{L}_{\mathrm{GAN}} + \lambda_{\mathrm{rgb}}\mathcal{L}_{\mathrm{rgb}} + \lambda_{\mathrm{lpips}}\mathcal{L}_{\mathrm{lpips}}, \tag{4}$$

where $\lambda_*$ is the loss weight of each item.

### 3.2 FRAMEWORK OVERVIEW OF UNCONDITIONAL GAN DISTILLATION

Knowledge distillation is a common strategy that can bring improvements in classification tasks. However, in generation tasks, its prerequisite, namely the student and teacher having consistent outputs for the same input, is rarely mentioned. In the absence of this prerequisite, the influence of mimicking losses to the training of student remains largely unknown. Here, we hypothesize

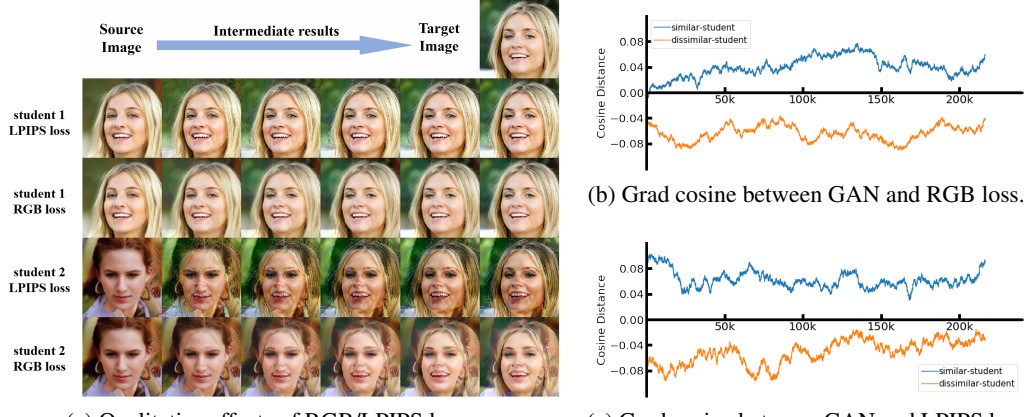

(a) Qualitative effects of RGB/LPIPS losses.

(b) Grad cosine between GAN and RGB loss.

(c) Grad cosine between GAN and LPIPS loss

Figure 1: (a) Student-1 has similar outputs with teacher and Student-2 has different outputs. The left column is the generated images of the original student. The image in the top right corner is the teacher output. We demonstrate the intermediate results to show how RGB/LPIPS loss influences the image generation of student. (b)(c) Cosine distance between the gradient of GAN loss and RGB/LPIPS loss. The $x-$axis denotes training steps. For similar student, RGB/LPIPS loss is cooperating with GAN loss. For dissimilar student, RGB/LPIPS loss is competing with GAN loss.

that RGB or LPIPS loss is not compatible with GAN loss when the output discrepancy occurs and distillation will also bring no benefit to the student. We examine this hypothesis both qualitatively and quantitatively.

Note that the three losses in Eq. 4 serve different roles. $\mathcal{L}_{\text{GAN}}$ requires the student to generate realistic images while $\mathcal{L}_{\text{rgb}}$ and $\mathcal{L}_{\text{lpips}}$ encourage similarity between the generated images by student and those of the teacher. Intuitively, if the generated image of the student are totally different from that of teacher for the same input, $\mathcal{L}_{\text{rgb}}$ and $\mathcal{L}_{\text{lpips}}$ will result in images that are slightly closer to teacher but with much less realism. To examine this hypothesis, we remove the GAN loss in Eq. 4 and keep only RGB or LPIPS loss. We also cut off the gradient backward path between the student generator and generated images. In this condition, the gradient of RGB/LPIPS loss directly works on the images. The change of synthesized images reflects how RGB/LPIPS loss influences the generation process. We select two student models, i.e., student-1 that has similar output with teacher for the same input and student-2 that has totally different outputs from teacher. The effects of RGB/LPIPS loss are shown in Fig. 1a. We can find that the intermediate results are a mixup of source and target images to some extent. If the source image is in the neighbor of the target image (1st and 2nd rows), the intermediate results are still perceptually realistic. However, if the source image has a large distance to the target image (3rd and 4th rows), the intermediate results are no longer realistic. Though RGB and LPIPS losses are reducing the distance between source and target images, they cannot guarantee a smooth and face-like interpolation in dissimilar setting. And this unrealistic intermediate results naturally contradict with GAN loss.

From quantitative perspective, we wish to prove that RGB/LPIPS loss is not compatible with GAN loss in the heterogeneous setting by gradient analysis. In the training process, for each batch, we perform backward propagation for GAN loss, RGB loss and LPIPS loss, respectively, and obtain three gradients of these losses. We then compute the cosine distance between GAN gradient and RGB/LPIPS gradients. As shown in Fig. 1b and Fig. 1c, the cosine distance between GAN gradient and RGB/LPIPS gradients of dissimilar student is always negative, suggesting that RGB/LPIPS gradients are competing with GAN gradients. On the contrary, the cosine distance of similar student is positive, indicating that the distillation loss is driving the model in the same direction as the adversarial loss. Our analysis above suggests that distillation is not beneficial in heterogeneous setting. Having similar outputs for the same input $z$ is the prerequisite for uncGAN distillation.

## 3.3 EFFECT OF THE STYLE MODULE

As we will show in the experiments, if the student is randomly initialized, it cannot learn consistent outputs as teacher even though we leverage RGB/LPIPS loss to force the agreement between the

outputs of two models. We hypothesize that the style module $S(z)$ plays a key role in determining whether two models can have consistent outputs. If the gap between style modules of student and teacher is too large, it is hard for the student to learn outputs consistent with the teacher.

We prove this hypothesis using a proof by contradiction. Suppose the student has a different style module from the teacher and the consistency condition (Eq. 2) is also satisfied. For the convenience of the following discussion, we define:

$$G(z_1, z_2; k) = C(w_1, w_2; k) = C(w_1, \cdots, w_1, w_2, w_1, \cdots, w_1), \quad (5)$$

where all the ModConv layers use $w_1$ except that the $k$-th layer uses $w_2$. The consistency condition of Eq. 2 requires that:

$$G_s(z_1, z_2; k) = G_t(z_1, z_2; k), 1 \le k \le L. \quad (6)$$

As shown in StyleGAN (Karras et al., 2019), for a well-trained model, the $w$ latent space consists of linear subspaces. It should be possible to find direction vectors that consistently correspond to individual factors of variation. An example is shown in Fig. 3. Some individual semantic factors such as pose, glasses and hair color can be controlled by moving the style vector $w$ of certain layer along a certain direction. Suppose the direction $p$ at $k$-th layer controls the hair color of the generated face. The only difference between $C_t(w_0, w_1 + p; k)$ and $C_t(w_0, w_1; k)$ is that they are the same faces with different hair colors. The movement of $w$ from $w_1$ to $w_1 + p$ corresponds to a consecutive change of hair color of the generated face. If we map the $w$ back to the noise space:

$$z_1 = S_t^{-1}(w_1), \quad z_2 = S_t^{-1}(w_1 + p), \quad (7)$$

obviously, the line segment in $w$ space corresponds to a curve in $z$ space with two end points $z_1$ and $z_2$ due to the nonlinearity of $S_t(z)$. We denote this curve as $\widehat{z_1 z_2}$. Then $\{G_t(z_0, z; k) | z \in \widehat{z_1 z_2}\}$ represents a cluster of faces with different hair colors. According to the consistency constraint, $\{G_s(z_0, z; k) | z \in \widehat{z_1 z_2}\}$ should be the same cluster as $\{G_t(z_0, z; k) | z \in \widehat{z_1 z_2}\}$. We feed $z_0, z \in \widehat{z_1 z_2}$ into the student style module $S_s(\cdot)$:

$$w_0' = S_s(z_0), \quad \widehat{w_1' w_2'} = S_s(\widehat{z_1 z_2}). \quad (8)$$

Since $S_s(\cdot)$ is different from and independent of $S_t(\cdot)$, the result $\widehat{w_1' w_2'}$ is still a curve. Thus, the semantic factor of hair color in student model is controlled by a complex curve in $w$ space, which contradicts the property of StyleGAN2 that various semantic factors are decoupled well in $w$ space. Hence, having different style modules and consistency condition cannot hold at the same time.

We further conduct experiments to examine our hypothesis. Specifically, we select four students according to whether the style module is from teacher or not and whether the convolution is from teacher or not. We use GAN loss, RGB loss and LPIPS loss to train these models. The style module and convolution are updated together. The results are shown in Fig. 2. The student that inherits weights from the teacher's style module can learn a mapping that aligns well with the teacher's output, no matter how the convolution $C(\cdot)$ is initialized. However, for the student whose style module is randomly initialized, there are no meaningful connections between student's and teacher's outputs. The analysis above clearly shows that the output consistency between student and teacher is determined by the style module.

### 3.4 STYLE MODULE CONSISTENCY IN GAN DISTILLATION

We have shown that the consistency between student and teacher outputs is the prerequisite of the distillation, and the style module determines whether two generators can have consistent outputs. Hence, to make distillation meaningful, it is necessary to impose extra constraints to guarantee the consistency between two style modules.

The simplest way is to keep the architecture of the style module unchanged and inherit teacher's parameters directly. In fact, the parameters of style module account for only 7.5% of the parameters of the convolution backbone. The FLOPs of style module account for about 0.005% of the FLOPs of the convolution backbone. Preserving style module is thus feasible in practice.

If there is a strong demand on the compression of style module, one can perform a two-stage training to ensure a small gap between student and teacher style modules. In the first stage, the student style module is forced to mimic outputs of the teacher style module:

$$\mathcal{L} = \mathbb{E}_{z \sim \mathcal{N}(0,1)} D(S_s(z), S_t(z)), \quad (9)$$

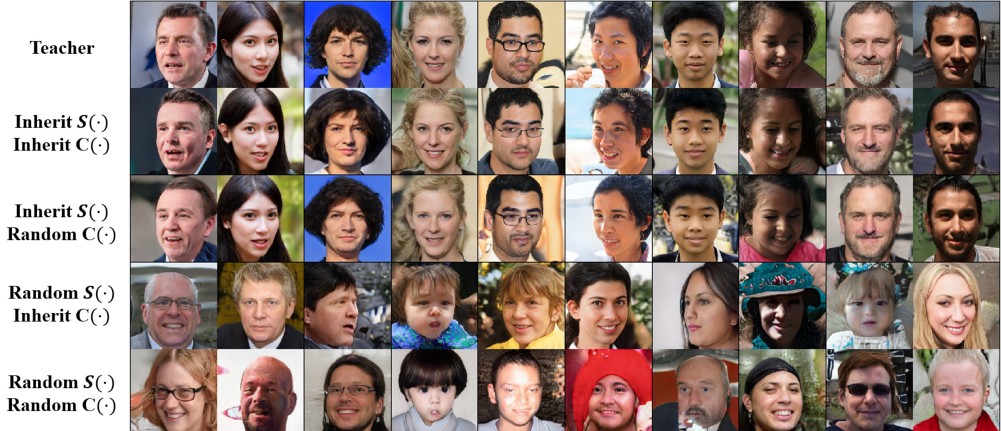

Figure 2: The style module $S(\cdot)$ determines whether student can learn from the teacher's output. If student inherits teacher's $S(z)$, it can learn the teacher's output well no matter how the convolution $C(\cdot)$ is initialized. If student uses random $S(z)$, it cannot learn the teacher's output no matter how the convolution $C(\cdot)$ is initialized.

where $D(\cdot, \cdot)$ is a distance metric. Considering $S_t(\cdot)$ and $S_s(\cdot)$ are both shallow MLPs, the training cost of this stage is negligible (0.59% of the normal GAN training in the second stage).

In the second stage, the style module and generator backbone are finetuned together using the loss in Eq. 4. We will explore the effects of compressing the style module in Sec. 4.1.

### 3.5 LATENT-DIRECTION-BASED RELATION DISTILLATION

Under the premise that consistency condition is satisfied, we further propose to incorporate relation mimicking into GAN distillation. Conventional relation-based distillation (Tung & Mori, 2019) in classification task computes feature similarity matrices using the samples in a minibatch. Here, we tailor it to better cater to StyleGAN2.

Specifically, for a given well-trained teacher model, we compute all its meaningful latent directions (LD) that control a single semantic factor and store them in a dictionary $\{d_1, d_2, \cdots, d_m\}$. Note that the latent direction is related to a specific layer. For example, if $d_i$ is computed in $k$-layer, then only $C_t(w, w + d_i; k)$ has single semantic factor difference with $C_t(w)$. $C_t(w, w + d_i; j)_{j \neq k}$ does not has this property. In the training stage, we feed a batch of noise $\{z_i\}_{i=1:N}$ into the style module and obtain $\{w_i\}_{i=1:N}$. For each $w_i$ we randomly sample a latent direction $d_j$ from the dictionary. Thus, $C_t(w_i)$ and $C_t(w_i, w_i + \alpha d_j; k)$ ($k$ is the layer related to $d_j$) are two images with single semantic factor difference with $\alpha$ controls the moving distance. We denote the intermediate features of $C_t(w_i)$ and $C_t(w_i, w_i + \alpha d_j; k)$ as $f_i$ and $f'_i$, respectively. Then the similarity matrix $M$ between original view and augmentation view can be computed as $A_{i,j} = f_i \cdot f'_j$. We then convert the similarity into probability via the softmax operation and minimize the distance using KL-divergence loss:

$$M_{i,j} = \frac{\exp(A_{i,j})}{\sum_{k=1}^{N} \exp(A_{i,k})}, \quad \mathcal{L}_{\text{LD}} = -\sum_{i,j} M_{i,j}^t \log M_{i,j}^s. \tag{10}$$

The final learning objective is the combination of Eq. 4 and $\mathcal{L}_{\text{LD}}$.

## 4 EXPERIMENTS

We conduct experiments mainly on StyleGAN2 since it is the most powerful unconditional GAN so far. We use the FFHQ (Karras et al., 2019) dataset. Following CAGAN (Liu et al., 2021), we adopt Fréchet Inception Distance (FID), Perceptual Path Length (PPL) (Karras et al., 2019) and PSNR/LPIPS (Liu et al., 2021) between real and projected images as evaluation metrics. More qualitative results are shown in the appendix.

Table 1: Effect of initialization. Inheriting both style module and convolution naturally obtains better results than random initialization. Surprisingly, we find that inheriting only style module is the best solution. With this initialization, RGB+LPIPS and RGB+LPIPS+LD obtains 0.38 and 0.51 improvements, respectively, compared to their corresponding results when inherit both modules.

| Style Module Initialization | Convolution Initialization | Mimicking Loss | Student FID |
|---|---|---|---|
| random | random | No Mimic | 10.92 |
| | | RGB | 10.78 |
| | | RGB + LPIPS | 11.27 |
| random | inherit | RGB | 10.81 |
| | | RGB+LPIPS | 10.88 |
| inherit | inherit | No Mimic | 10.54 |
| | | RGB | 9.41 |
| | | RGB + LPIPS | 8.61 |
| | | RGB + LPIPS + LD | 8.45 |
| inherit | random | RGB | 9.42 |
| | | RGB + LPIPS | 8.23 |
| | | RGB + LPIPS + LD | **7.94** |

For the ablation study in Sec. 4.1, we train the models on resolution 256×256 and use a smaller batch size of 8 to save the computation cost. For the comparison with state-of-the-art methods in Sec. 4.2, we train the models on both resolutions of 256×256 and 1024×1024. We also use a batch size of 16 that is the same as CAGAN (Liu et al., 2021) to ensure a fair comparison.

## 4.1 ABLATION STUDY

**The Initialization of the Student Model.** Previous works usually treat StyleGAN2 as an integral module and initialize style module and convolution backbone in the same way (from scratch or inherits teacher parameters). Based on our analysis in Sec. 3.3 that the style module plays a key role in determining the semantics of generated images, here we separate style module $S(z)$ from convolution backbone $C(w)$ and test three initialization strategies: 1) both $S(z)$ and $C(w)$ are randomly initialized, 2) both $S(z)$ and $C(w)$ are initialized with teacher weights, 3) only $S(z)$ inherits teacher weights and $C(w)$ is randomly initialized.

The results are shown in Table. 1. For the setting where $S(z)$ and $C(w)$ are both randomly initialized, RGB loss can only bring marginal improvement. RGB+LPIPS even performs worse than No-Mimic, indicating that distillation cannot work well when output discrepancy occurs. If $S(z)$ and $C(w)$ both inherit teacher weights, the mimicking loss can achieve 1-2 FID improvement. To explore the effect of style module, we also try inheriting only $S(z)$ and surprisingly find that this initialization obtains the best result. And loading $C(w)$ hampers the performance of RGB+LPIPS and RGB+LPIPS+LD. This result contradicts with the conclusion in CAGAN (Liu et al., 2021). It shows that the general pruning strategy, i.e., determining which channels should be removed, is not important. Randomly initialization of convolution layers is the optimal solution as long as the style module is kept.

**The Effects of Style Module Compression.** We conduct experiments to investigate how to deal with the style module in StyleGAN2 compression. Specifically, we consider three settings: 1) student has the same style module architecture as teacher but with random initialization, 2) student style module has a different architecture and uses the two-stage training strategy, 3) student has the same architecture and inherits the weights from the teacher style module. For all the settings, the convolution backbones are randomly initialized. For the two-stage setting, we also explore how the architecture of the style module and mimicking loss in Eq. 9 affect the final performance. To emphasize the importance of style module, we also list the average L1 distance between $S_s(z)$ and $S_t(z)$ before entering the normal GAN training stage.

The results are shown in Table. 2. 'Random Initialization' obtains the worst FID because the output discrepancy makes the distillation ineffective. The 'Two-Stage' strategy improves the results by narrowing the gap between $S_s(z)$ and $S_t(z)$. From several two-stage settings, we can find that L1

Table 2: How to deal with style module in StyleGAN2 distillation. The style module is comprised of MLPs. The numbers inside and outside the "[]" is the number of channels in each layer and the number of layers, respectively. The style module of the teacher is [512]*8. The FLOPs saving only considers FLOPs reduction in the style module.

| Setting | Style Module Architecture | FLOPs Saving | $D(\cdot, \cdot)$ | $\|\|S_s(z) - S_t(z)\|\|_1$ | Student FID |
|---|---|---|---|---|---|
| Random Initialization | [512]*8 | 0% | N/A | 1.027 | 11.78 |
| Two-Stage | [512]*8 | 0% | L1 | 0.156 | 9.69 |
| | [512]*8 | 0% | L2 | 0.260 | 10.80 |
| | [512]*5 | 37.5% | L1 | 0.197 | 10.38 |
| | [390]*7+[512] | 37.44% | L1 | 0.210 | 10.55 |
| | [256]*7+[512] | 68.75% | L1 | 0.245 | 10.86 |
| Inheriting | [512]*8 | 0% | N/A | 0 | **8.30** |

is a better mimicking loss than L2 and reducing the number of layers is better than reducing the number of channels in each layer. It is also worth noting that there is a strong positive correlation between $|S_s(z) - S_t(z)|$ and FID, indicating that the gap between $S_s(z)$ and $S_t(z)$ determines the output consistency and further determines the influence of distillation.

Though the two-stage strategy brings performance gains, there is still a large gap between it and the 'Inheriting' variant. Thus, we conclude that the modification to the style module will greatly harm the final performance and the two-stage strategy can only mitigate the degradation to a certain degree. Considering that the scale of the original $S_t(z)$ is negligible compared to the convolution backbone, the best practice in StyleGAN2 compression is to preserve the style module architecture and inherit the weights from the teacher style module.

**The Effects of Latent-Direction-Based Distillation Loss.** The proposed latent-direction-based loss is essentially a relation loss. We are interested in whether the benefit brought by $\mathcal{L}_{LD}$ comes from relation mimicking or from the latent-direction-based augmentation.

Table 3: Ablation study about relation mimicking. Single View brings marginal improvement. Random Offset even has negative effect. Our LD loss consistently improves the performances of both RGB and RGB+LPIPS.

| Mimicking Loss | $\mathcal{L}_{LD}$ | FID |
|---|---|---|
| RGB | N/A | 9.41 |
| RGB + Random Offset | KL | 9.80 |
| RGB + Single View | KL | 9.47 |
| RGB + LD | L2 | 9.16 |
| RGB + LD | KL | 9.05 |
| RGB + LPIPS | N/A | 8.61 |
| RGB + LPIPS + LD | L2 | 8.64 |
| RGB + LPIPS + LD | KL | **8.26** |

Specifically, we consider three variants: 1) Single View, namely the similarity is computed inside the normal samples rather than between normal samples and augmented samples, 2) Random Offset, namely we move $w$ along a random direction to get $f_i'$ instead of along the latent direction, 3) Our latent-direction-based method (abbreviated as LD).

The results are shown in Table. 3. RGB+Single View obtains comparable results with RGB loss, indicating that naive relation mimicking cannot bring improvements. RGB+Random Offset performs even worse than RGB loss, demonstrating that the way of augmentation is critical for relation mimicking. Improper augmentation may be counterproductive. LD improves RGB by around 0.3 FID. And LD can still bring 0.16 FID improvement to the RGB+LPIPS variant. Among two kinds of $\mathcal{L}_{LD}$, KL-divergence loss is slightly better than L2 loss.

## 4.2 COMPARISON WITH STATE-OF-THE-ART METHODS

We compare our method with the GAN Slimming (Wang et al., 2020) and CAGAN (Liu et al., 2021) methods. Since our method does not focus on the pruning, we directly adopt the student architecture used in CAGAN, i.e., a network that is the same as teacher but with fewer channels. We also compare with CAGAN in heterogeneous setting where the student is not a subnet of the teacher. Specifically, we modify the kernel size of the second convolution layer in each resolution block from 3 to 1, thus inheriting teacher convolution parameters is infeasible. Since CAGAN did not notice the output

Table 4: Comparison with SOTA. "↓" ("↑") denotes the lower (higher) the better. For the FID and PPL of Full-Size, GAN slimming and CAGAN, we directly use the results reported in CAGAN (Liu et al., 2021). Since CAGAN did not release their computation details of PSNR and LPIPS, we compute these two metrics using our own implementation. We also list the PSNR and LPIPS results reported in CAGAN in "()". The PSNR and LPIPS of GAN slimming are blank due to the lack of its well-trained checkpoint. **Bold** font denotes the results that outperform CAGAN. "heter" denotes heterogeneous setting where the student is not a subnet of the teacher.

| Model | Image Size | FLOPs | FID (↓) | PPL (↓) | PSNR (↑) | LPIPS (↓) |
|---|---|---|---|---|---|---|
| Original Full-Size | 256 | 45.1B | 4.5 | 0.162 | 34.26 (32.02) | 0.057 (0.113) |
| Baseline | 256 | 4.1B | 9.79 | 0.156 | 33.14 | 0.082 |
| GAN slimming | 256 | 5.0B | 12.4 | 0.313 | (31.02) | (0.177) |
| CAGAN | 256 | 4.1B | 7.9 | 0.143 | 33.34 (31.41) | 0.076 (0.144) |
| Ours | 256 | 4.1B | **7.25** | **0.135** | **33.49** | **0.071** |
| CAGAN-heter | 256 | 2.7B | 13.75 | 0.158 | 33.19 | 0.083 |
| Ours-heter | 256 | 2.7B | **9.96** | **0.141** | **33.54** | **0.073** |
| Original Full-Size | 1024 | 74.3B | 2.7 | 0.162 | 33.52 (31.38) | 0.075 (0.149) |
| GAN slimming | 1024 | 23.9B | 10.1 | 0.211 | (30.74) | (0.189) |
| CAGAN | 1024 | 7.0B | 7.6 | 0.157 | 32.63 (30.96) | 0.099 (0.170) |
| Ours | 1024 | 7.0B | **7.19** | **0.128** | **32.70** | **0.094** |

discrepancy issue and always initialize the style module and convolution backbone in the same way, we assume it does not inherit weights from teacher in heterogeneous setting. The results are shown in Table 4. Our method outperforms CAGAN on FID by 0.65 and 0.41 on resolution 256×256 and 1024×1024, respectively, showing that our method can generate more realistic images. Note that these improvements are not marginal considering the images generated by CAGAN already have high quality. For the PPL metric that measures the smooth degree of latent space, we outperform CAGAN by 0.008 on resolution 256. The gap is even larger (0.029) on resolution 1024. For PSNR and LPIPS that are related to image projection ability, our method also surpasses CAGAN, demonstrating that our method can model the face distribution in real world better. Our superiority is much more significant in heterogeneous setting, showing that our method can be applied in a more general situation where the student is not necessary to be a subnet of the teacher.

## 5 CONCLUSION

In this paper, we uncover the output discrepancy issue in uncGAN distillation. Through comparative experiments, we find that the style module is the key to the output discrepancy and propose a novel initialization strategy of student, which can help resolve the output discrepancy issue. The proposed latent-direction-based distillation loss further improves the distillation efficacy and we achieve state-of-the-art results in StyleGAN2 distillation, outperforming the rival method by a large margin on image realism, latent space smoothness and image projection fidelity.

## 6 LIMITATIONS

The analysis and method in this work are tailored for StyleGAN2-like generator, which may affect its universality. It is unknown whether our analysis is still valid for those uncGANs that do not have style module such as BigGAN (Brock et al., 2019) and PGGAN (Karras et al., 2018). Whether conventional mimicking loss such as RGB/LPIPS loss can solve the output discrepancy and how to initialize them are not clear. Besides, we mainly conduct experiments on FFHQ which is a well-aligned face dataset. Experiments on more and harder datasets such as ImageNet, outdoor scenes and rooms should be included to examine the effectiveness of our method. Finally, we only consider the output discrepancy issue in unconditional GANs. In fact, this problem also exists in conditional setting when the condition is not strong enough (e.g., the conditional input is the class label). How to analyze the output discrepancy issues of uncGANs and cGANs in a more general form is also a direction worth exploring.

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

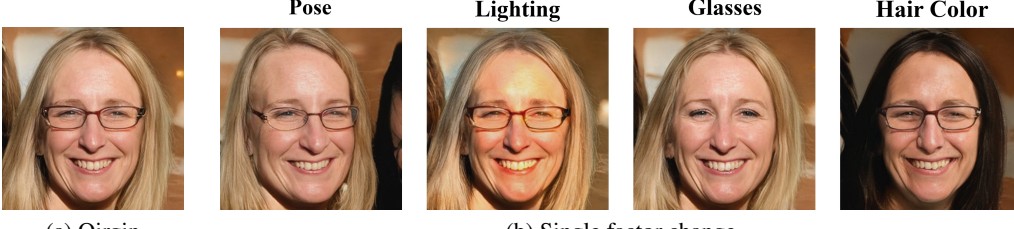

| Pose | Lighting | Glasses | Hair Color |

(a) Oirgin            (b) Single factor change

Figure 3: StyleGAN2 shows good factorization in the $w$ space. It is possible to control a single semantic factor such as pose, lighting condition, glasses and hair color by moving the style vector $w$ of a certain layer along a specific direction.

APPENDIX

## A  IMPLEMENTATION DETAILS

**Training Hyperparameters.** For the style module mimicking (Eq. 9) of the first stage, we use Adam optimizer with a initial learning rate of 0.05. We train for 50k steps with a batch size 4096. For the normal GAN training of the second stage, we use Adam optimizer with a initial learning rate of 0.002 and 450k iterations. For the $\alpha$ that controls the offset along latent direction, we sample it from a Gaussian distribution $\mathcal{N}(0, 5)$. We set $\lambda_{\mathrm{GAN}}$, $\lambda_{\mathrm{rgb}}$, $\lambda_{\mathrm{lpips}}$ and $\lambda_{\mathrm{LD}}$ to be 1, 3, 3 and 30, respectively.

**Evaluation Metrics.** Fréchet Inception Distance (FID) is commonly used metric to evaluate the realism of generated images. The generated images and real images are fed into a inception network and then a Fréchet distance is computed between their corresponding feature maps. We use the implementation of FID in CAGAN (Liu et al., 2021). Specifically, we use 50K real images and 50K generated images to compute statistics, respectively. Perceptual Path Length (PPL) is proposed in StyleGAN (Karras et al., 2019) to measure the smoothness of latent space. We adopt the PPL implementation in CAGAN (Liu et al., 2021) for a fair comparison. PSNR and LPIPS are proposed by CAGAN to evaluate the image projection ability. A given real image is first mapped back to the latent space through optimizer such as L-BFGS. The projected image is obtained by feeding this resulting latent code to the generator. Then the PSNR and LPIPS distance are computed between projected image and the original image again. A smaller value indicate that the generator can model the distribution in real world better. We compute these two metrics using our own implementation.

## B  STYLEGAN2 LINEAR SEPARABILITY

A well-trained StyleGAN2 model is linear separably in latent space. An example is shown in Figure. 3.

## C  IMAGE EDITING

We demonstrate the superiority of our method on image editing, including style mixing and interpolation. Given two real face images $I_A, I_B$, we first project them back to the latent space and get $w_A, w_B$. Both $w_A$ and $w_B$ are of shape $L \times D$, where $L$ is the number of convolution layers and $D$ is the dimension of latent code. For style mixing, we replace the $i-$th vector in $w_A$ with that from $w_B$. We set $i \in [1, 3]$, $i \in [5, 8]$ and $i \in [10, 13]$ for coarse, middle and fine style mixing, respectively. For interpolation, we linearly combine the latent code with $\beta$ controls the weight: $w = \beta \cdot w_A + (1 - \beta) \cdot w_B$, and then feed $w$ into generator to get the interpolation results. We edit the images on resolution 256×256.

The results are shown in Fig. 4. For style mixing, CAGAN always has artifacts in face shape (coarse style) and skin color (middle shape). While our results are more realistic and correspond better with two source images. In coarse style case, our result corresponds well on face shape and sense organs with source B. In fine style case, our result corresponds well on lighting and skin color with source

B. For interpolation, we also observe a smoother change than CAGAN, showing that our method learns a better structure in latent space.

## D    IMAGE PROJECTION

We show image projection results of our method in Fig. 5. All the real images come from Helen Set55 (Liu et al., 2021) and are not seen in the training stage. Our model reconstructs them with high quality.

## E    GENERATION RESULTS

We show more generation results of resolution 256×256 and resolution 1024×1024 in Fig. 6 and Fig. 7, respectively.

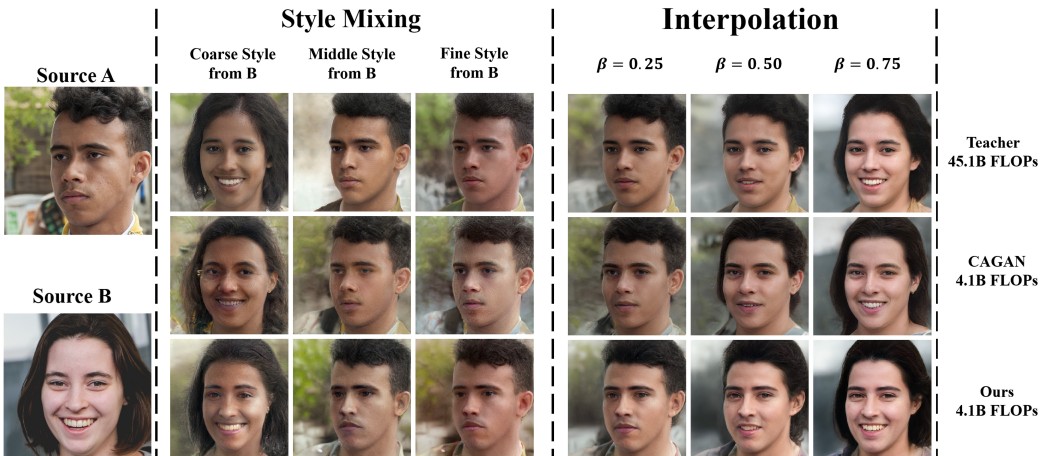

(a) In coarse style mixing, CAGAN generates glasses, which does not appear in both source images. CAGAN also produces blurry images in middle style case. In contrast, our style mixing results are more realistic and more similar to teacher. Our method also shows smoother transition of mouth and haircut in interpolation.

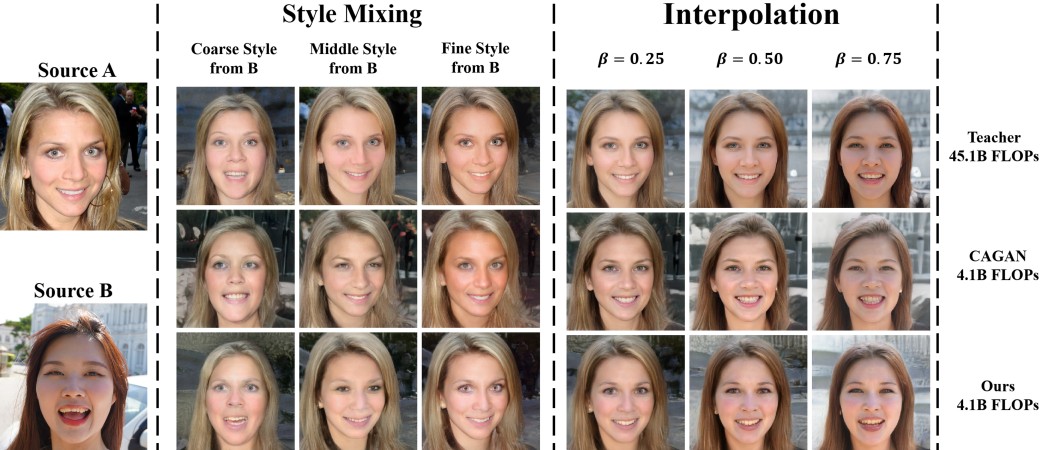

(b) In coarse style mixing, our result corresponds better with source B on mouth and face shape. In fine style mixing, our result corresponds better with source B on skin color. CAGAN also generates artifacts on hair in middle and fine style cases. In interpolation, our methods are more consistent with teacher than CAGAN.

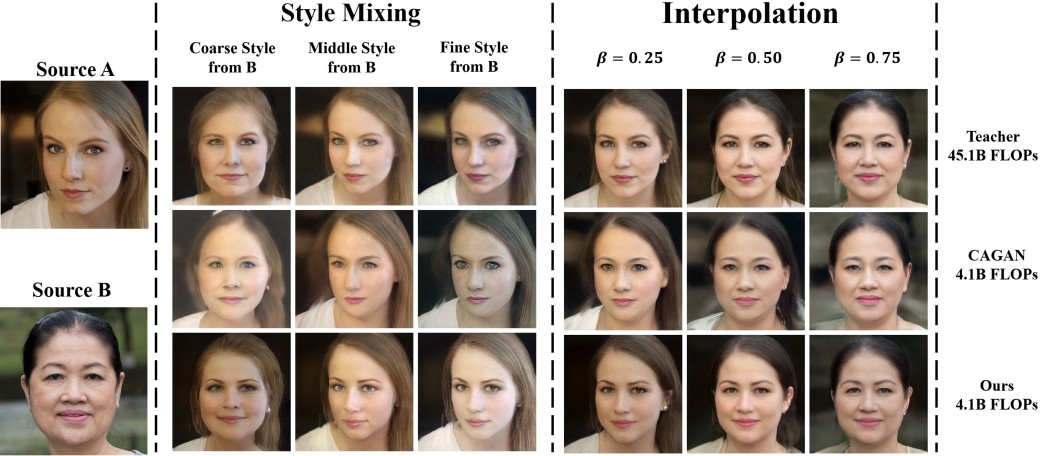

(c) CAGAN generates lighting artifacts in coarse case and skin color artifacts in fine case, while our results are more realistic. In interpolation of CAGAN, the earrings disappear in $\beta = 0.25$ while appear again in $\beta = 0.50$. In contrast, our results are much smoother.

Figure 4: Image editing results.

**Real** **Proj.** **Real** **Proj.** **Real** **Proj.** **Real** **Proj.**

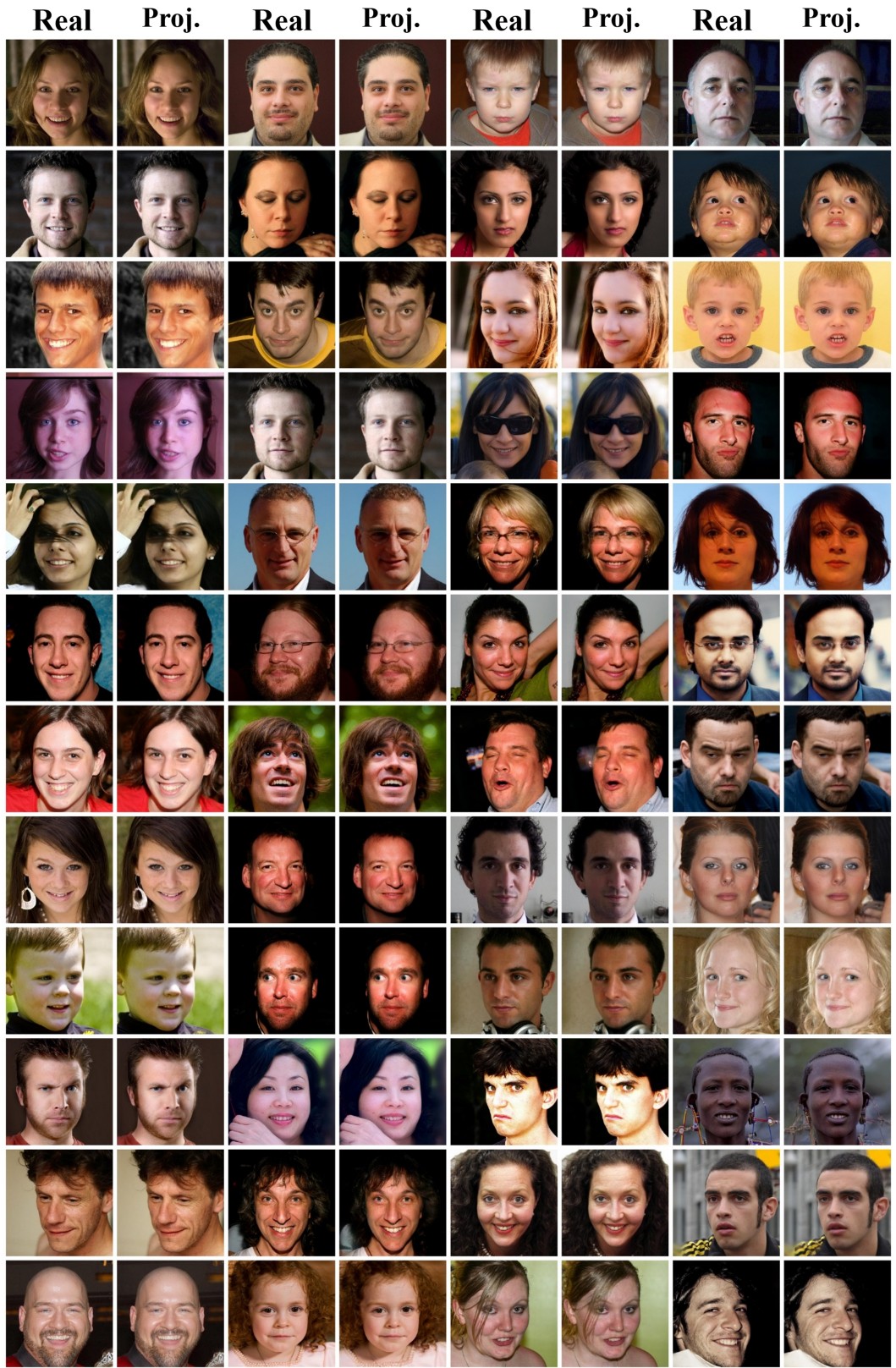

Figure 5: Image projection results. In each pair, the left image is from real world (not from training set) and the right image is the projected result by our model. Our method can model the real face distribution well.

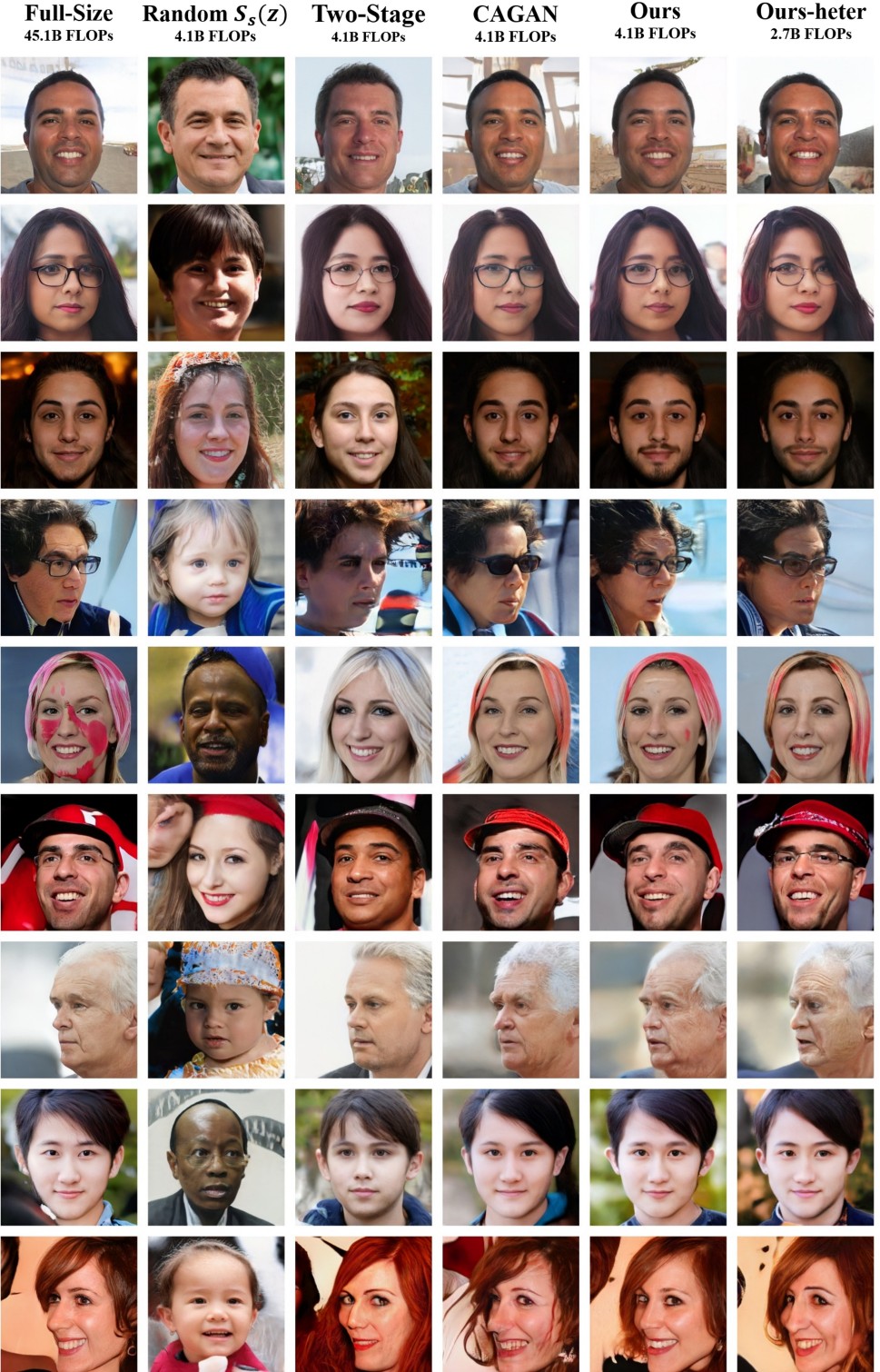

Figure 6: Synthesized results on resolution 256×256. Random $S_s(z)$ denotes the style module $S_s(z)$ is randomly initialized. For Two-Stage, we compress the original 8-layer style module into 5 layers. The images of each row are generated using the same input noise $z$. Note that all the students are trained with mimicking loss. Random $S_s(z)$ cannot generate images consistent with teacher due to the different style module. Two-Stage mitigate output discrepancy issue by directly mimicking style module, but there are still semantics differences from teacher. Compared to CAGAN, our generated images have fewer artifacts and are more similar to teacher.

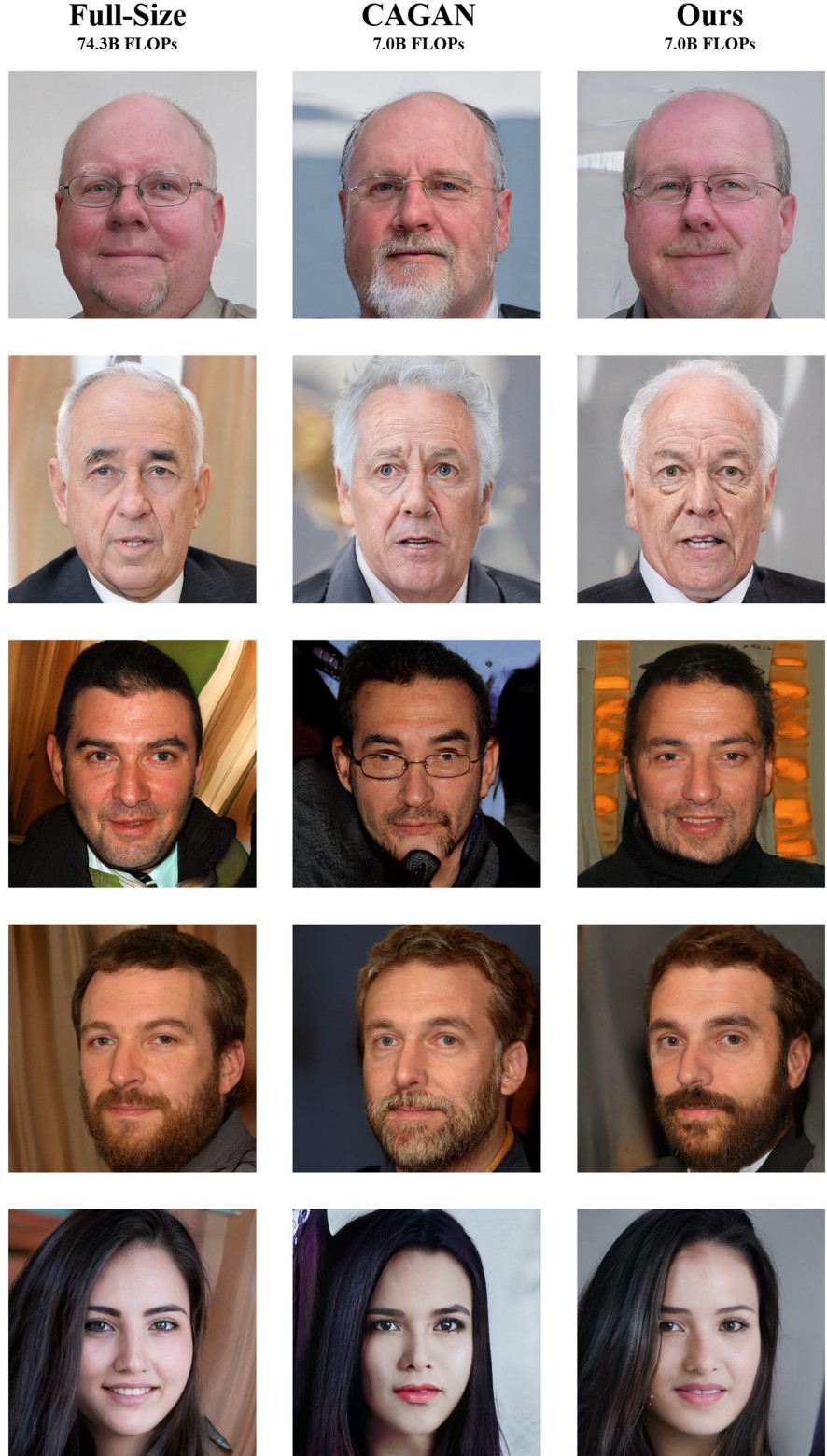

Figure 7: Generation results on resolution 1024×1024. The synthesized images of Our method are of better quality than CAGAN. In several semantic factors such as beard, haircut and glasses, our results are more similar to the full-size model even though we do not inherit convolution weights.

