# OpenReview forum: "A Comprehensive Overhaul of Distilling Unconditional GANs"
_ICLR.cc/2022/Conference — ICLR 2022 Submitted_

### Official Review · Reviewer_MFc1 · 2021-10-30

**Correctness:** 4
**Technical Novelty And Significance:** 3
**Empirical Novelty And Significance:** 2
**Recommendation:** 6
**Confidence:** 3

**Main Review:**

Strengths:
- This could be a good study paper for the unconditional GAN. The authors conducted several investigations and show the effect of the style module. Both the proposed student initialization and the latent-direction-based distillation are new and interesting for the unconditional GAN distillation.
- The proposed method achieves better performance than the recent SOTA, including GAN-slimming and CAGAN, on FFHQ dataset.
- The proposed initialization is well-motivated and very straightforward. Also, the paper is well-written and easy to follow.

Weakness:
- To keep the final performance, the style module architecture cannot be changed. Although style module is pretty light weighted, it is a potential limitation.
- Most experiments are conducted on human face domain (FFHQ dataset). In the current form, it is unclear whether the proposed method can work well for other domains, like animals, or scenery images.

**Summary Of The Paper:**

The authors firstly conduct investigations and show that it is important to make sure the consistency between the style modules of both teacher and student networks. Based on their observation, a simple yet effective initialization is proposed -- initialize the student style module with that of the teacher one. The authors further compute the latent directions and use that for augmentation. Overall the proposed method is well-motivated and interesting. The proposed method outperforms GAN-slimming and CAGAN.

**Summary Of The Review:**

I think it is a good candidate that could bring contributions to our community. Although the topic is a bit narrow to unconditional GAN, the experiments are thorough. Most of their claimed are supported by their experiments.

--------------------------- After Rebuttal ---------------------------


Thank you for the feedback. It is good to see more experiments such as the new results on LSUN church dataset. The authors have addressed my questions. However, I agree with other reviewers that the contribution is a bit narrow (or limited). I think this paper is really borderline, and I have no objection for acceptance/rejection.

---

> ### Author Response · Authors · 2021-11-21
> **Response to Reviewer MFc1**
>
> Q1: To keep the final performance, the style module architecture cannot be changed. Although style module is pretty light weighted, it is a potential limitation
>
> A1: The style module can be changed. There is a trade-off between the change to style module and final performance. From the perspective of model compression, i.e., achieving high performance with small model, we recommend to keep the style module unchanged. But When compressing the style module becomes a rigid demand, the proposed two-stage training strategy is an alternative.
>
> Q2: Most experiments are conducted on human face domain (FFHQ dataset). In the current form, it is unclear whether the proposed method can work well for other domains, like animals, or scenery images.
>
> A2: We add extra experiments on LSUN church dataset. See response to #R1, Q3. We will also conduct experiments on more datasets in the future.

---

### Official Review · Reviewer_RUSD · 2021-11-02

**Correctness:** 3
**Technical Novelty And Significance:** 2
**Empirical Novelty And Significance:** 2
**Recommendation:** 5
**Confidence:** 5

**Main Review:**

**Pros:**

1. The output discrepancy issue pointed out by the paper is a real problem for distillation, which does not only exist in the StyleGAN2 compression (references [1] and [2]). The valuable conclusion is that this paper provides the solutions for the StyleGAN2 compression. Based on their results, we know that the style module can not be compressed, while CNN backbone can be compressed. Because the style module of the student should inherit from the teacher.

2. The illustrated proof and result are convincing. Section 3.2 shows the following two phenomena: 1) The more similar output between studentNet and teacherNet makes studentNet generate more realistic images. 2) The dissimilar outputs cause GAN loss and distillation loss to have a competitive relationship, thereby affecting the distillation.

3. This paper provides comprehensive experiments, including both qualitative analysis and quantitative results, to show the effectiveness of the proposed framework. The paper is well written, and the authors analyze the reasons step by step and propose reasonable solutions.

[1] J. Han, et al. Fixing the Teacher-Student Knowledge Discrepancy in Distillation.

[2] B. Liu, et al. MetaDistiller: Network Self-Boosting via Meta-Learned Top-Down Distillation.


**Cons:**

1. The paper didn’t introduce the difference between the image generation task and recognition task in the knowledge discrepancy problem, and the inspiration of the proposed method on the use of knowledge distillation in other fields. Otherwise, the proposed method is too specific.

2. Based on the results of the references [1] and [2], the output discrepancy issue also exists in the classification task, although there is enough supervised loss for the training of studentNet. Thus, it seems that the above issue also exists in the uncGANs distillation, which is contrary to the authors’ claim in the 3rd paragraph of the introduction. In order to prove the correctness of the hypothesis in the paper, there should be detailed theoretical analysis or experimental results.
In section 3.3, the counter-evidence provided by the authors is simple and not convincing. Of course, a well-trained Net and random initialized Net generate different latent space W, and the W of the latter Net is certainly not linearly separable. And, for two different sizes of style modules, the conclusion is the same as above.

3. Insufficient comparative experiments. 1) For Figure 3, there are no results for different initialized CNN backbone. Without them, how can one demonstrate that the style module plays a more important role in the distillation?  And in Table 1, what about the results on the random-inherit setting. 2) The experimental results of tuning hyperparameters should be added in the paper (\lambda_GAN, \lambda_rgb, \lambda_lpips, and \lambda_LD).  The authors should provide more details of them under different settings, whether they are with the same values in the results of Table 2.

4. Previous work [3] proposed the initialization of studentNet for compressing GAN. This paper claims that the randomly initialized studentGAN can not learn well from teacherGAN. Thus, they provide a two-step training approach. There are some concerns: 1) Since there is a competitive relationship between GAN loss and distillation loss in the early training stage, what happens if you gradually reduce the weight of GAN loss for training (studentGAN with random initialization)? In this case, the style module could also be compressed instead of directly inheriting from teacherNet.

[3] Z. Zhang, et al. P-KDGAN: Progressive knowledge distillation with GANs for one-class novelty detection.


**Minor comments:**

1. Table 4 seems not very clear to me. It has many horizontal lines to divide different experimental results, which makes it difficult to read.


**Questions during rebuttal period:**

1. Please address and clarify the cons above.

2. In table 1, how do you explain the results, 9.41 vs. 9.42. It seems not compatible with the paper’s claim. In addition, are these numbers the average of several experiments?
Is the competitive relationship between GAN loss and distillation loss not alleviated in the late training phase? Please provide results similar to Figure 1 (c).


**Some typos:**

1. Section 4.2: 0.65 and 0.31 -> 0.65 and 0.41


**Summary Of The Paper:**

The paper identifies a key factor that affects the knowledge distillation on unconditional GANs, in particular the StyleGAN2 model: the output discrepancy between teacherNet and studentNet. It proposes that the initialization of the StyleGAN2’ style module causes the above problem to be more serious. To verify their claims, the authors show detailed analysis and comparative experiments. They also propose a novel latent-direction-based loss to further improve the distillation.

**Summary Of The Review:**

Please see the content in the main review. I tend to recommend ICLR 2022 to reject this paper. If my concerns can be well solved, I will increase my score.


--------------------------- After Rebuttal ---------------------------

Thanks for your detailed feedback.

I totally agree with the authors that the discrepancy problem is different between classification and unconditional generation. The discussion in the introduction and results in Figure 2 and Table 1 can well demonstrate their claims.

The results in Q4 should be added in the paper.

However, as other reviewers said, the contribution is limited. The proposed compression method is only verified in one model, StyleGANv2. Thus, I encourage the authors to explore more modules to improve the influence.

---

> ### Author Response · Authors · 2021-11-21
> **Response to Reviewer RUSD**
>
> Q1: didn’t introduce the difference between generation task and recognition task in the discrepancy problem ...
>
> A1: The output discrepancy is a general concept. If two models cannot produce identical outputs, it can be seemed as output discrepancy. Hence, it exists in both generation task and recognition task. However, the discrepancy degree is not the same. In the recognition task, the groundtruth label guarantees that two models can naturally converge to a similar mapping. Take a 2-category classification task as a toy example, the teacher may output a distribution (0.90, 0.10) and the student may output a distribution (0.86, 0.14). Though these two distributions are not identical, the discrepancy is small. The distillation loss will push the student distribution to teacher distribution. However, in unconditional GAN task, for the same input noise, a teacher may generate a face and student may generate an apple. RGB/LPIPS loss cannot guarantee a smooth and realistic interpolation from apple to face, resulting a competition with GAN loss.
> In this work, we focus on the large discrepancy problem in unconditional GANs. It is not suitable to be applied in recognition task. Considering that uncGAN is a broad term and StyleGAN2 is the most powerful uncGAN, we believe that our work has its value.
>
> Q2: ... contrary to the authors’ claim in the 3rd paragraph ...
>
> A2: We explain the difference between discrepancies in classification task and generation task in Q1. And it does not contradict our statement in the 3rd paragraph of Intro.
> We have proven our hypothesis that having similar style modules is the necessary condition of having similar outputs from both theoretical and experimental perspectives.
> 1) Theoretical perspective. We wish to point that no matter how the style module is initialized (even randomly), the style module will be linearly separable after StyleGAN-like training paradigm. This property is the base of our proof by contradiction. In Sec. 3.3, we show that if two models have similar outputs but with different style modules, then at least one of them is not linearly separable, which contradicts the linear separability of StyleGAN2. Thus, having similar outputs and having different style modules cannot hold at the same time.
> 2) Experimental perspective. As shown in Figure 3, we use two students with one inheriting teacher style module and one using random initialization. Though the style module here is fixed, we have also tried updating style module jointly. No matter whether style module is updated, the random initialized student cannot learn teacher output. This further indicate the rightness of our hypothesis.
>
> Q3: Insufficient comparative experiments
>
> A3: We add the experiment where the style module is randomly initialized, and convolution is from teacher. See the results in Table 1 in revised version. We can find that inheriting convolution or not does not change the FID significantly if style module is randomly initialized. We also update the qualitative results in Figure 3. It is clear that the style module determines the discrepancy to teacher.
>
> For the loss weights of GAN, RGB and LPIPS, we directly adopt the values in CAGAN. For the LD loss weight, we tried 10,30,60 and100. Other training hyperparameters are the same as that in Table 4. The FID results are 7.36, 7.25, 7.31 and 7.42. Hence, we finally set the loss weight to be 30.
>
> Q4: ...gradually reduce the weight of GAN loss for training ...
>
> A4: In fact, we have tried several loss schemes:
>
> 1) distillation loss + gradually reducing GAN loss,
> 2) distillation loss + gradually increasing GAN los,
> 3) gradually reducing distillation loss+ GAN loss,
> 4) gradually increasing distillation loss + GAN loss
>
> These variants give the same results, i.e., student cannot generate similar outputs to teacher. And their FID are all around 10.9 (baseline in Table 1). So far, we have not found an effective way to solve the discrepancy for a random initialized student.
>
> Q5: In table 1, ..., 9.41 vs. 9.42. ...
>
> A5: Due to the limited computation resources, we only conduct repeated experiments for the CAGAN and our method on resolution 256 in Table 4 (see response to #R2, Q1). The numbers in Table 1 are the results of single-trial experiment.
> In this paper, we claim that initializing the style module and convolution with teacher weights and random weights, respectively, is a better initialization manner. This claim is based on the global performances under two initialization settings. In Table 1, though our method only achieves comparable result on RGB (9.42 vs. 9.41), it brings large improvements on RGB+LPIPS (8.23 vs. 8.61) and RGB+LPIPS+LD (8.45 vs. 7.94).
> The competitive relationship between GAN loss and distillation loss only exists in the case where student style module is randomly initialized. In this case, student cannot generate similar outputs to teacher in the whole training process. The competition is still not alleviated in the late phase.

---

### Official Review · Reviewer_F96C · 2021-11-03

**Correctness:** 2
**Technical Novelty And Significance:** 2
**Empirical Novelty And Significance:** 2
**Recommendation:** 5
**Confidence:** 4

**Main Review:**

#### Strengths:
* important topic (generative model compression)
* the paper is relatively easy to follow
* state of the art model considered together with a standard dataset in the experimental section

#### Weaknesses:
* the novelty appears incremental and is very tailored to a single architecture (StyleGAN2)
* the validation is not convincing (a single state of the art architecture is considered and results are reported on a single dataset for this architecture), improvements seem marginal
* limitations of the proposed approach are not discussed

#### Details on the above-mentioned concerns:
The technical novelty of this paper appears somewhat overstated. The proposed initialization seems borrowed from the pruning-based literature in GAN compression, as stated in the paper "Note that pruning is only applied to the convolution backbone C and the style module S is kept unchanged". If the style module remains identical to the one used in the pruning-based literature, then the randomly initialization of the convolutional layers of the student network would remain the potential benefit of the proposed approach, as one would have presumably more options to consider in terms of architecture design. Moreover, although the paper argues rather strongly against pixel-wise losses and perceptual losses in the unconditional GAN setting, those losses are still preserved in the model leading to the best performance (perhaps based on the observation on the importance of the style module). The paper also introduces a loss term inspired by the relational distillation introduced in previous work. While the proposed loss appears sound, the improvements it brings seem rather small. Unfortunately, there is no standard deviations reported in the empirical results, making it hard to assess whether those apparently small improvements are significant.

* Conditional GANs is a very broad term, and whereas output discrepancy may be a limited problem in some settings (e.g. the discussed image to image translation task), other settings will likely experience diverse outputs given some condition (e.g. class conditional GANs can generate diverse outputs given a class label). The paper should be more clear wrt the use of this terminology.
* "... if the generated image of the student are totally different from that of teacher for the same input, Lrgb and Lpips will result in images that are slightly closer to teacher but with much less realism." Perhaps this is a more philosophical question, but from the perspective of model compression, should we expect student/teacher networks to output the same images given a noise vector (student network emulating the mapping learnt by the teacher network) or should we expect the student network to be able to mimic the distribution of images of the teacher network? If the former, the experiments that assume that teacher and student will result in very different generations should be better motivated, why should we care about this scenario? If the latter, it is unclear why the paper does not consider distilling knowledge via a discriminator loss between student and teacher generations.
* The solution is very tailored to StyleGAN2. It would be worth exploring other state of the art generative models. For example, it has been shown that StyleGAN2-like architectures do not generalize well to large scale and diverse datasets such as ImageNet. What about leveraging BigGAN in an unconditional scenario? (see e.g. https://arxiv.org/abs/1903.02271, https://arxiv.org/abs/2012.02162, https://arxiv.org/abs/2109.05070 for unconditional BigGAN options)
* The experimental validation could be strengthened by considering additional datasets (e.g. ImageNet or COCO-Stuff).
* Also, what happens if the unconditional problem is transformed into a conditional one where the condition comes from e.g. clustering the data points (see e.g. https://arxiv.org/abs/1709.07359, https://arxiv.org/abs/1712.04407, https://arxiv.org/abs/2006.10728)?
* "Specifically, we select two students with one inheriting teacher’s style module and the other being randomly initialized for the style module. The convolution backbones are both randomly initialized. We keep the style modules of two students frozen and only train their remaining convolution layers." Is this the case for the results reported in the rest of the paper? It would seem natural to allow the style module which is randomly initialized to be updated during training. How would that compare to the 2 stage training in terms of results?
* Limitations of the work should be acknowledged and discussed.

#### Additional comments:
* Table 1 does not show the full picture. It would be beneficial to include and discuss results for the missing configurations (e.g. LPIPS is never considered on its own, LPIPS + LD is not considered, RGB+LD is only sometimes considered, etc).
* Although the paper argues that improvements are not marginal, it would be best to assess their significance. Please include std to all results, given the small improvements reported, it is very hard to assess the significance of the results.
* It would be beneficial to report the teacher FID in all tables to ground the discussion further.
* In Table 2, it is unclear why report l1 between s(z) for teacher/student since this same quantity is what appears to be optimized in the two stage training. Therefore, it would not appear to be surprising that the gap is narrowed for the 2 stage training when using L1 as distance metric.
* It is good practice to explain how FID was computed: What was the reference data used? How many samples (both from the reference and generated images) were used to compute the metric?
* In Table 4, LPIPS the higher the better (arrow is upside down).
* Please double check bolding in Table 4, sometimes "ours" results are bolded but CAGAN achieves slightly higher performance (e.g. LPIPS).
* Given the topic of the paper, it would seem appropriate to have smaller tables and gaining space to include student/teacher generations side to side for a grid of randomly selected images.



**Summary Of The Paper:**

This paper tackles the problem of image unconditional generative model compression by leveraging the knowledge distillation framework. The paper highlights some challenges of applying KD out of the box on unconditional generative models and proposes to copy a subset of pretrained parameters (style modules) directly from the teacher to the student network. The paper further proposes an additional loss term, which preserves semantic relations in the latent spaces. Experiments are conducted using a StyleGAN2 backbone on the FFHQ dataset.

**Summary Of The Review:**

Overall, although the topic covered in this paper is relevant to the ICLR community, the reviewer does not find the paper ready for acceptance. In particular, the contributions seem rather narrow and the empirical validation is not compelling. For details, see the main review.

**Post-rebuttal update:** The reviewer thanks the authors for the feedback which addressed some of their concerns. However, some the reviewer's concerns wrt the stated contributions remain. More precisely, claiming the initialization strategy (already adopted in pruning methods) as contribution appears a bit stretched, and the compression approach being tailored specifically to StyleGANv2 makes the contribution rather narrow. There might be value in the analysis and lessons learnt wrt compressing StyleGANv2, but considering the limited results and their unknown generalizability to other models and more challenging datasets, the reviewer considers that the paper is not ready for acceptance.

---

> ### Author Response · Authors · 2021-11-21
> **Response to Reviewer F96C**
>
> Q1: The technical novelty of this paper appears somewhat overstated. The proposed initialization seems borrowed from the pruning-based literature in GAN compression ...
>
> A1: It is true that pruning is applied to the convolution layers and the style module is kept unchanged. However, the most common practice in pruning literature is to inherit parameters from both convolution and style module, because the convolution after pruning is still a subnet of the original network. Hence, our proposed method is different from the common solution in pruning-based literature.
> Although our final method seems simple and natural, the value of analysis behind this method cannot be ignored. In the Sec. 3.2 and Sec. 3.3, we provide thorough analysis about the effect of style module. It serves as ground evidence to support our method. Without this analysis, initializing C and S in different manner just because C is changed and S is unchanged, will be a totally intuitive method.
>
>
> Q2: Moreover, although the paper argues rather strongly against pixel-wise losses ...
>
> A2: We do not argue strongly against RGB or LPIPS loss. In fact,  these two losses are both useful if the discrepancy issue can be solved.
> We rerun and update the result of RGB+LPIPS+LD variant in Table 3. Due to the time limit, we did not run repeated experiments for the variant in Table 3. However, according to the result in Table 4 (see response to #R2, Q1), the std level is around 0.1. Thus, the improvements brought by LD (0.36 and 0.35) are not marginal.
>
> Q3: The paper should be more clear wrt the use of this terminology.
>
> A3: Thanks for the comments. In our first submission, we regard output discrepancy issue as a unique property of uncGAN and cGAN does not have this problem. As you suggest, cGAN can also output diverse images if the condition is not strong enough (e.g., category label). We have revised out statement to a more precise one in Sec. 1.
>
> Q4: ...should we expect student/teacher networks to output the same images given a noise vector....
>
> A4:  Both these two manners are reasonable in KD. In our first submission, we only considered the first possibility and did not provide discussion or reason. We have added it in the Sec. 1 in revised version.
>
> Q5: The solution is very tailored to StyleGAN2
>
> A5: We acknowledge that this work is tailored to StyleGAN2 and not applicable to other GAN algorithms. However, considering that StyleGAN2-like generator is still the most powerful generator till now, we think this work has its value.
>
>
> Q6: The experimental validation could be strengthened by considering additional datasets
>
> A6: Due to the time limit of rebuttal, we only add experiments on LSUN church dataset (see response to #R1, Q3). The experiments on ImageNet and BigGAN will be our future work direction.
>
> Q7: Also, what happens if the unconditional problem is transformed into a conditional one ...
>
> A7: We believe that if the condition in cGAN is weak (e.g. category label, from human annotation or self-supervised clutering), the output discrepancy issue still holds. We will conduct experiments about weak-condition GAN in the future.
>
>
> Q8: ... It would seem natural to allow the style module which is randomly initialized to be updated during training ...
>
> A8: The quantitative results of this setting are not reported in the rest of the paper, because we just intend to show that the style module is the necessary condition to solve output discrepancy issue. For all the experiments in Sec. 4, the style module is normally updated no matter how it is initialized.
>
> Q9: Limitations of the work should be acknowledged and discussed.
>
> A9: We have added limitation sections in the revised version.
>
> Response to additional comments:
>
> For all the experiments in Table 1/2/3, we use the same teacher with FID 4.5.
>
> In Table 2, we are not going to show that the two-stage strategy can reduce the L1 gap of S(z). We intend to show that there is a strong correlation between this gap and the final student FID. The smaller the gap is, the smaller the student FID is. Thus, the best solution is to directly inherit teacher style module.
>
> For the computation of FID, we add them in appendix A in revised version.
>
> The LPIPS is indeed the lower the better, because it measures the distance between real image and projected image. The detailed explanation can be found in CAGAN paper Sec. 4.1. And our bold LPIPS results in Table 4 do represent the best results.

---

### Official Review · Reviewer_bbKQ · 2021-11-04

**Correctness:** 2
**Technical Novelty And Significance:** 2
**Empirical Novelty And Significance:** 2
**Recommendation:** 3
**Confidence:** 4

**Main Review:**

Pros:
A- The paper tackles an interesting problem
B- The authors are referring and comparing to key previous work (CAGAN).


Cons:
C- My biggest concern with the current draft is the experiment section and claims. While the authors have done a good job in comparing to previous work (CAGAN), the comparison is not convincing nor rigourous. Here is why:
1) Standard deviation is missing in all tables. The authors need to run several train for each experiment  and report their average and standard deviation. All numbers between CAGAN and the proposed method are so close that a standard deviation is required to compare these methods. It is a common practice in GAN literature. I am quite surprise that the authors have omitted it.
2) The authors claim (in the conclusion) that they outperform previous work by a
“large margin”. It is not supported by any evidence. I am not going to list all the   quantitative numbers but when the gains are “0.008 or 0.029 etc…” on a metric where previous work had one order of magnitude more gain, to my humble opinion, these are not large.
3) While the quantitative results can not be interpreted without the standard deviation,  qualitatively comparison are also not convincing.

D- Finally, the technical depth of the paper and contributions are also limited. I am in favour of simple yet effective ideas. Yet, they need to be “effective”. Here, following my comments in the experiment section, it seems that the ideas are simple but not necessarily effective.

Here are some typos to be fixed:
- “sduent” -> student in Sec 2.
- “0.31” -> 0.41 in Section 4.2

**Summary Of The Paper:**

The authors present a method to distill StyleGAN2, an unconditional Generative adversarial network. They claim that the main challenge lies in the output discrepancy between the teacher and student. They have identified that the style module plays an important role in determining semantic information of generated images. Hence, they have proposed:
i) to initialise the student model with the inherent weights of the teacher module and keeping the remaining convolutional layers randomly initialised, and
ii) a new distillation loss that preserves the semantic relations in latent space.
The authors run all their experiment on StyleGAN2.

**Summary Of The Review:**

Given C and D comments, I can not accept the paper “as is”. Critical points are missing and conclusions can not be drawn without them. I hope that the authors will be able to address my comments. I thank them for their work and highly encourage them to continue this line of work and re-submit. Best.

---

> ### Author Response · Authors · 2021-11-21
> **Response to Reviewer bbKQ**
>
> Q1: Standard deviation is missing in all tables
>
> A1: Since the training of StyleGAN2 on resolution 1024 is a time-consuming process (>500 GPU hours), we only conduct repeated experiments on resolution 256. Specifically, we run CAGAN and our method for 3 times. The results are 7.9±0.08 and 7.25±0.13, respectively. The improvement 0.65 is greatly larger than the fluctuation, showing the effectiveness of our method.
>
> Q2: ...“large margin”. It is not supported by any evidence.
>
> A2: First, we should consider the relative improvement. The mentioned “0.008 or 0.029” are both from PPL metric. PPL itself is already a small value. Thus, our actual relative improvements are 5.6% and 18.5%, which are not marginal.
> Besides, it seems that the improvement of our method (compared to CAGAN) is much smaller than the improvement of CAGAN (compared to GAN slimming). We wish to point that the result of GAN slimming is from CAGAN paper and is worse than expectation, making the gap between CAGAN and GAN slimming seem large. In our experiments, we surprisingly find that the baseline (from scratch, without any compression technique) has already achieved much better results than GAN slimming. We have added baseline results in Table 4. When we make an overall consideration of baseline, CAGAN and our method, we think our improvement is not marginal.

---

### Official Review · Reviewer_fZmP · 2021-11-05

**Correctness:** 3
**Technical Novelty And Significance:** 3
**Empirical Novelty And Significance:** 3
**Recommendation:** 6
**Confidence:** 4

**Main Review:**

Strength:
  1. Solving the output discrepancy issue for distilling unconditional GAN is an interesting topic.
  2. Show many ablation studies for different initialization strategies and loss functions for style module and CNN parts.

Concerns:
  1. Since the paper abstract starts with the motivation of reducing the storage and computation of deploying GAN models, I would like to see a thorough comparison between a distilled model using the proposed method and a same architecture network which is simply trained from the scratch using the original training recipe described in StyleGAN paper. Otherwise, if a same architecture can achieve the same quality by training from scratch, there is no need to do distilling any more. When I read the paper, I guessed the row 1 in Table 1 (random + random + No Mimic) might represent such experiment, but there is no discussion about how this experiment is done.  It also needs to be added to Table 4.
  2. In Section 3.2 and Figure 1(a), it is unclear to me how to find the similar student models and dissimilar student models. What type of criteria is used to define similar or not similar?
  3. The paper only shows the results on face images (the aligned FFHQ face images). How does the method work on other types of images? Especially, other types of images, like building/rooms, have more complicate data distribution than aligned face images. The semantic information of these types of images are not that well disentangled as face images. In this case, does inheriting style module still help? Does the latent-direction-based distilling loss still help?
  4. In addition to RBG+LPIPS+LD losses, would adding adversarial loss also help?

**Summary Of The Paper:**

Typical knowledge distillation loss/pipeline failed for unconditional GAN distillation, due to the output discrepancy between teacher and student model even if same inputs are fed. This paper proposed a framework for distilling unconditional GANs. The framework mainly contains two parts: 1). Inherit the weights from teacher style module for initializing student’s style module based on the observation that style module plays a crucial role in determining the outputs’ semantic information; 2). Propose a latent-direction-based distillation loss to enforce the consistency between teacher’s outputs and student’s outputs. Experiments are mainly shown by comparing with CAGAN and some variants of the proposed methods with different initialization strategies and loss functions.

**Summary Of The Review:**

Overall, the paper tries to address an interesting problem of tackling the output discrepancy issues when distilling unconditional GAN models. The experiment part provides several ablation studies. However, I have several main concerns regarding to: 1). more thorough comparison with the same architecture network trained from scratched and more discussions; 2). try the proposed method on other types of images to validate whether inheriting style modules and latent-direction-based distilling loss still help or not.


==============After Rebuttal===========================

Thanks the authors for the clarification. I agree with other authors about the limitations of the proposed method, especially about how to extend to other unconditional GANs and other datasets. Thus, I will still give a borderline rating.

---

> ### Author Response · Authors · 2021-11-21
> **Response to Reviewer fZmP**
>
> Q1: Results of baseline
>
> A1: The variant “random + random + No Mimic”  in Table 1 is indeed the baseline experiment. However, the goal of Table 1 is to examine the influence of initialization, thus we did not refer it as baseline. Table 4 is comparing various GAN compression methods, and we have added the baseline results in Table 4 for a better understanding of our improvement.
>
> Q2: How to find the similar student models and dissimilar student models. What type of criteria is used to define similar or not similar?
>
> A2: In Figure 1(a), the similar student is obtained by inheriting teacher’s style module and the dissimilar student is obtained by initializing style module randomly. The current definition of being similar or not is subjective, i.e., relying on human judgement. However, we wish to emphasize that the goal of Figure 1(a) is just to illustrate the different intermediate results of two students and our algorithm does not require a strict and objective definition of similarity. Hence, a subjective definition can already satisfy our need.
>
>
> Q3: Results on other datasets
>
> A3: We conduct extra experiments on LSUN Church dataset. However, due to the tight time limit of rebuttal, we cannot transfer all the experiments (e.g., ablation study) from FFHQ to new dataset. Hence, we select several important experiments to demonstrate the effectiveness of our method on LSUN Church. Specifically, the teacher, baseline, CAGAN and our method achieves 9.21, 15.37, 12.85 and 12.02 FID, respectively.
>
>
> Q4: Would adding adversarial loss also help?
>
> A4: The “adversarial loss” here is a little vague. Indeed, all the experiments have already incorporated original GAN loss. So, we guess the adversarial loss here refers an adversarial mimicking manner. There is another discriminator classifying whether the generated images are from teacher or student, thus making the student learn teacher’s distribution. We believe this loss can bring improvement.

---

### Decision · Program_Chairs · 2022-01-20

**Decision:**

Reject

**Comment:**

The paper proposes a method for compressing unconditional generative models by leveraging a knowledge distillation framework. Two reviewers consider the paper slightly above the acceptance threshold for the interesting topic studied in the paper and the simplicity of the method. However, the other three reviewers consider the paper below the acceptance threshold with two reviewers rating the paper slighting below the acceptance threshold and one reviewer rating the paper as not good enough. Several issues were raised, including that the paper only contains results from one unconditional model (StyleGAN2) and that the presented results are not convincing enough. Consolidating the reviews and the rebuttal, the meta-reviewer found the concern raised by the reviewers justified. It would be more ideal if the paper can present results on different unconditional models and more datasets. The authors are encouraged to incorporate the reviewers' feedback to make the paper stronger for a future venue.